# Evidence for Quercetin as a Dietary Supplement for the Treatment of Cardio-Metabolic Diseases in Pregnancy: A Review in Rodent Models

**DOI:** 10.3390/foods11182772

**Published:** 2022-09-08

**Authors:** Paulo César Trindade da Costa, Evandro Leite de Souza, Diego Cabral Lacerda, José Patrocínio Ribeiro Cruz Neto, Ludmilla Christine Silva de Sales, Cristiane Cosmo Silva Luis, Paula Brielle Pontes, Marinaldo Pacífico Cavalcanti Neto, José Luiz de Brito Alves

**Affiliations:** 1Postgraduation Program in Nutrition Sciences, Health Sciences Center, Federal University of Paraiba, João Pessoa 58051-900, Brazil; 2Postgraduation Program in Physiological Sciences, Federal University of Paraíba, João Pessoa 58051-900, Brazil; 3Postgraduation Program in Neuropsychiatry and Health Sciences Behavior, Federal University of Pernambuco, Recife 50670-901, Brazil; 4Integrated Laboratory of Morphofunctional Sciences, Institute of Biodiversity and Sustainability (NUPEM), Federal University of Rio de Janeiro, Macaé 21941-901, Brazil

**Keywords:** polyphenols, quercetin, pregnancy complications, metabolic diseases

## Abstract

Quercetin supplementation during pregnancy and lactation has been linked to a lower risk of maternal cardio-metabolic disorders such as gestational diabetes mellitus (GDM), dyslipidemia, preeclampsia, attenuation of malnutrition-related conditions, and gestational obesity in animal studies. Pre-clinical studies have shown that maternal supplementation with quercetin reduces cardio-metabolic diseases in dams and rodents’ offspring, emphasizing its role in modifying phenotypic plasticity. In this sense, it could be inferred that quercetin administration during pregnancy and lactation is a viable strategy for changing cardio-metabolic parameters throughout life. Epigenetic mechanisms affecting the AMP-activated protein kinase (AMPK), nuclear factor-kappa B (NF-κB), and phosphoinositide 3-kinase (PI3 K) pathways could be associated with these changes. To highlight these discoveries, this review outlines the understanding from animal studies investigations about quercetin supplementation and its capacity to prevent or decrease maternal and offspring cardio-metabolic illnesses and associated comorbidities.

## 1. Introduction

Experimental studies in animals and epidemiological findings have demonstrated that metabolic disorders experienced in utero or early life are risk factors for the permanent changes in the structure and function of organs and systems in dams and offspring. According to the developmental origins of adult health and disease (DOHaD) hypothesis, early environmental insults, such as nutritional deficits, lead to maladaptive responses in the uterus and placenta, which affect fetal growth during critical developmental windows and increase the risk of cardio-metabolic diseases in adulthood [1,2].

Gestational diabetes mellitus (GDM) and hypertensive disturbances, such as preeclampsia, are cardio-metabolic illnesses associated with enhanced mother and progeny mortality. Furthermore, malnutrition and gestational obesity are risk factors for maternal-fetal complications. It has been demonstrated that offspring born from mothers displaying cardio-metabolic disorders during pregnancy exhibit a high risk of obesity and diabetes mellitus in adulthood [3,4]. This is partly due to oxidative stress, increased pro-inflammatory cytokines, and poor adipokine signaling [3]. In this sense, antioxidant and anti-inflammatory interventions may be essential in treating maternal cardio-metabolic diseases and preventing harmful consequences in offspring.

Polyphenols are phytochemicals found mainly in fruits and vegetables and are subdivided into flavonoids, stilbenes, phenolic acids, and lignans [5]. Polyphenol-rich foods or dietary supplements have been associated with a lower risk of cardiovascular disease and body weight management in undernutrition and gestational obesity [6,7,8].

Quercetin is a flavonoid-related polyphenol linked to a lower risk of cardiovascular disease in preclinical studies. There is growing interest in studying quercetin’s antioxidant and anti-inflammatory properties in various pathological conditions, particularly chronic non-communicable diseases [5]. However, the efficacy of quercetin supplementation in maternal cardio-metabolic and associated disorders remains unclear.

In this sense, it is plausible that quercetin supplementation during pregnancy and lactation could be considered a nutritional approach to reduce the risk of maternal and offspring cardio-metabolic disorders and related causative conditions [9,10,11]. Given its therapeutic potential in pregnant rat and mice, as demonstrated by early studies, this review focuses on new insights into the effects and benefits of quercetin intervention on both maternal metabolic disorders and offspring health [12,13]. Animal evidence on the effects of quercetin supplementation in maternal and offspring cardio-metabolic conditions will be discussed. To further explore the intervention efficacy, aspects related to dosage, duration of treatment, gestational time of exposure, and the main findings reported in the mother and offspring are also discussed.

## 2. Quercetin

Polyphenols are secondary metabolic products that are among the most abundant and widely distributed natural compounds obtained from plants. Polyphenols are found as simple molecules, such as phenolic acids, or as highly polymerized compounds, such as tannins [14]. Polyphenols are usually found in conjugated forms with one or more sugar residues connected to the hydroxyl group, in direct bonds with aromatic carbon or chemicals, such as carboxylic and organic acids, amines, lipids, and other phenols [15].

Several polyphenol structures have been identified, including phenolic acids, flavonoids (flavonols, flavanones, isoflavones, and anthocyanins), lignans, stilbenes, polyphenolic amides, and other polyphenols [5,16]. Dietary polyphenols are essential for human health mainly due to their antioxidant and anti-inflammatory properties, helping prevent and slow disease progression in subjects with metabolic syndrome, cardiovascular disease, diabetes mellitus, chronic degenerative diseases, and cancer [8,17].

Quercetin is a flavonol found in citrus fruits, apples, berries, onions, green tea, green leafy vegetables, seeds, nuts, broccoli, olive oil, grapes, and red wine [18]. Biotransformation enzymes acting on phase II metabolism produce quercetin metabolites, such as methylated, sulfated, and glucuronidated [19]. Anti-inflammatory, antioxidant, antidiabetic, antihypertensive, antiobesity, antihypercholesterolemic, antiatherosclerosis, anticancer, and antitumor activities have been reported as biological effects of quercetin [19].

Quercetin is found in functional foods in levels ranging from 10 to 120 mg per serving as well as in dietary supplements in doses ranging from 200 to 1000 mg per day [20]. The use of quercetin as a dietary supplement is generally considered safe for adults. However, there is no data from clinical studies in humans attesting to the safety of quercetin supplementation in pregnant or lactating women, although it has already been shown that such supplementation did not cause adverse effects on the reproductive system of pregnant animals [21].

Given that oxidative stress, inflammation, apoptosis, and endothelial dysfunction have been reported to predict cardio-metabolic diseases [22], evidence indicated the protective benefits of quercetin in animals exposed to maternal cardio-metabolic disorders [23,24]. Quercetin reduces malondialdehyde (MDA), reactive oxygen species (ROS), oxidized low-density lipoprotein (ox-LDL), protein carbonyl, and inducible nitric oxide synthase (iNOS) while increasing the expression of superoxide dismutase (SOD), catalase (CAT) and glutathione peroxidase (GPx). The improvement of inflammation occurs by reducing the expression of cyclooxygenase (COX), Arachidonate 5-lipoxygenase (5-LOX), C-reactive protein (CRP), monocyte chemoattractant protein (MCP), Toll-like receptors (TLRs), nuclear factor-kappa B (NF-κB), interleukin-6 (IL-6), interleukin-1B (IL-1B), tumor necrosis factor-α (TNF-α), intercellular adhesion molecule-1 (ICAM-1), and vascular cell adhesion molecule-1 (VCAM-1) [22].

## 3. Quercetin Intervention in Rodents during Pregnancy and Lactation

Studies have been developed to investigate the effect of some substances during pregnancy and lactation. Regarding quercetin, studies have shown that supplementation of this flavonol during these stages of life can have beneficial effects for both the mother and the fetus [25,26,27].

Pregnant mice who received quercetin (302 mg/kg) from 3 days before conception until the 14th gestational day showed no adverse effects on placental or fetal development. Furthermore, quercetin increased iron storage and decreased liver oxidative stress in mice by upregulating genes of the antioxidant defense system [25].

Quercetin, which has an estrogen-like chemical structure and a high affinity for the estrogen receptor ligand-binding domain, may have a role in the development of estrogen-related illnesses. The protective impact of quercetin on damaged endometrial cells in pregnant rats (5.5 days’ gestation) was tested at a concentration of 50 mol/L, alleviating injured endometrial cells by boosting the expression of CYP1A1 and CYP2B1 and lowering TNF-α and IL-6 levels [26].

There are reports of anomalies in fetuses from dams treated with theophylline during pregnancy, such as aortic displacement and ventricular septal defects [28]. The authors evaluated the effects of quercetin intervention on theophylline-induced abnormalities in pregnant Wistar rats that received 100 mg/kg of quercetin on the 9th and 10th days of pregnancy. In this study, quercetin prevented theophylline-induced cardiac anomalies in rat embryos, reduced MDA levels, and increased GPx activity [28], which indicates that some fetal abnormalities may result from increased free radicals and oxidative stress.

Using an LPS-induced experimental preterm labor model in mice, an early study demonstrated that quercetin administration at 30, 90, and 150 mg/kg administered on the 15th day of gestation reduced preterm labor by 63% and increased the offspring survival rate. This occurred through mechanisms involving NF-κB inhibition, a key transcription factor activated during a pro-inflammatory state [29].

During pregnancy and lactation, quercetin supplementation (50, 100, and 200 mg/kg) reduced oxidative stress, which was associated with increased activity of CAT and SOD enzymes as well as with decreases in pro-inflammatory cytokines interleukin-17 (IL-17) and interleukin-22 (IL-22) levels. Furthermore, quercetin supplementation improved the integrity of the intestinal barrier, as evidenced by upregulated expression of zonula occludens (ZO-1) and occludin as well as an improvement in the intestinal microbiota linked to the reduction in the relative abundance of *Bacteroides* [27].

Another study evaluated the effects of quercetin intervention in mice with postpartum hypogalactia. The authors observed that quercetin supplementation (50 mg/kg day) for 10 days postpartum improved mammary gland development and lactation yield, probably due to stimulating prolactin expression and release of the pituitary gland [30].

In summary, quercetin can exert beneficial effects during pregnancy and lactation associated with its antioxidants, anti-inflammatory, and antiobesity properties. The therapeutic potential of quercetin in attenuating maternal and offspring cardio-metabolic illnesses, including GDM, hypertensive syndromes in pregnancy, and causative factors, such as gestational undernutrition and obesity, are discussed in the following sections. The knowledge of data on the efficacy of quercetin in rodent models can allow an understanding of the potential of this bioactive molecule in preventing or rescuing signals and symptoms related to maternal cardio-metabolic disorders within a translational perspective.

## 4. Quercetin Intervention in Gestational Diabetes Mellitus

### 4.1. GDM: A Pregnant Disorder Linked to Oxidative Stress and Inflammation

GDM is the most common cardio-metabolic disorder that affects women during pregnancy, characterized as any degree of glucose intolerance with onset or first diagnostic during the gestational period and does not meet the diagnostic criteria for pre-existing diabetes mellitus [31]. Multiple etiological variables may contribute to this condition, including a family history of diabetes, maternal obesity, physical inactivity, and increasing maternal age. Hyperglycemia, gestational hypertension, higher birth weight, preterm birth, birth injury, neonatal hypoglycemia, neonatal unit admission, and respiratory distress are some negative impacts of GDM on mother and child throughout pregnancy and lactation [32]. Oxidative stress and low-grade inflammation have been implicated in the pathogenesis of GDM [31,33]. The NF-κB-dependent pathway links maternal hyperglycemia to increased expression of nitric oxide synthase 2 (NOS-2) and production of reactive nitrogen species [34,35]. This condition causes a stressful intracellular environment marked by nitrosative, oxidative, and endoplasmic reticulum stress, which triggers apoptotic mechanisms in the neural folds, leading to neural tube abnormalities in the embryo [36,37,38].

### 4.2. Quercetin Supplementation in Rodent Models of GDM

Animal evidence shows that quercetin supplementation can reduce GDM and related symptoms in dams and offspring (Table 1). To understand its therapeutic potential, the effects of quercetin on the regulation of nitric oxide synthase 2 were investigated in mice subjected to GDM induced by streptozotocin (STZ) [39]. Oral supplementation with quercetin (100 mg/kg) during neurulation (from embryonic day 6.5–9.5) reduced neural tube defects, apoptosis rate, nitric oxide synthase 2 (NOS-2) expression, and nitrosative stress parameters, followed by improvement of antioxidant responses by increased levels of superoxide dismutase 1 (SOD1) in STZ-induced-GDM mice. Furthermore, quercetin downregulated NOS-2 expression by modulating the NF-κB by reducing p65 levels, a family member (NF-κB), and increasing the κB kinase-α, which suppresses the p65 nuclear translocation. Quercetin also reduced NOS-2 gene expression and nitrosative stress in diabetic animal embryos, followed by decreased endoplasmic reticulum (ER) stress. These results indicated that quercetin treatment reduced the effects of hyperglycemia on the NF-κB pathway and restored them to pre-diabetic levels [39].

Quercetin supplementation (100 mg/kg) administered during the embryonic period E7.5 to E10.5 reduced the rates of neural tube defects (1.4%) in fetuses from pregnant diabetic mice, comparable to those from non-diabetic dams, while the untreated offspring had a defect rate of 23%. The treatment was also associated with apoptosis reduction in the dorsal neural tube, represented by lower TUNEL-positive apoptotic bodies and Caspase-3 cleavage compared to diabetic untreated mice. Furthermore, it was observed via high-performance liquid chromatography (HPLC) and electrospray ionization-tandem mass spectrometry (ESI-MS/MS) analysis that the effects of the treatment on the fetus would be associated with the action of secondary metabolites arising from the mother’s metabolization of quercetin, such as methylated and sulfonylated derivatives, which were transferred by maternal–embryonic interface and reached the embryo. In addition, quercetin supplementation reduced nitrosative stress, which was linked to lower levels of NOS-2 expression and protein nitrosylation and nitration, contributing to lowering embryonic and fetal abnormalities in diabetic animals. Moreover, quercetin reduced oxidative stress in the neural tissues of treated diabetic animals, causing an increase in the expression of antioxidant enzymes, such as SOD1 and superoxide dismutase 2 (SOD2), and a reduction in oxidative stress markers, including 4-hydroxynonenal and MDA, compared to untreated diabetics. Although quercetin induced multiple beneficial effects for embryo development, this biomolecule did not change maternal outcomes. In summary, these effects show the potential for this polyphenol and its metabolites to cross the placental barrier, contributing to reducing neural tube defects in the embryo induced by hyperglycemia [24].

The placenta is altered by STZ-induced diabetic embryopathy, increasing the number of glycogen cells and the thickness of the labyrinth interhemal membrane (LIM). The LIM is the closest point of circulation between the mother and the fetus, and it plays a crucial role in the exchange of chemicals between maternal and fetal blood, being the primary location of hemotrophic exchanges. LIM is considered a selectively penetrable trilaminar obstacle. Its expansion can increase the distance between maternal and fetal circulation, decreasing the supply of nutrients to the fetus. Quercetin supplementation at the dose of 5 mg/kg administered on days 0, 14, and 20 of pregnancy has been shown to reverse the adverse effects of GDM on the rat placenta, promoting a reduction in the number of glycogen cells and LIM thickness [41].

Quercetin increases placental adiponectin expression while decreasing AdipoR1 and AdipoR2 receptor expression. Adiponectin is an adipokine that helps prevent placental abnormalities in GDM by improving insulin sensitivity. Pregnant women with GDM have low serum adiponectin levels from the beginning through the end of pregnancy. Insulin resistance causes hyperglycemia, which raises the risk of postpartum type 2 diabetes, fatty liver, and cardiovascular disease. In conclusion, through modulating adiponectin and its receptor signaling, quercetin can enhance structural changes in the placenta [41].

Balouki et al. (2020) investigated the effect of quercetin on hyperglycemia-induced changes in developing preimplantation embryos from female mice with STZ-induced GDM. The researchers observed that 30 mg/kg daily of quercetin, starting from 4 weeks before pregnancy, effectively reduced maternal glucose levels, increased the number of embryos, and improved embryonic morphological distribution. In addition, quercetin increased the 17β-estradiol (primary estrogen) levels and enhanced the estradiol/progesterone ratio but did not change the progesterone level. Estrogen can directly affect preimplantation embryo development by stimulating insulin-like growth factor 1 (IGF1) production in the reproductive system. Quercetin ameliorated the diabetes-induced negative effect by increasing the expression of IGF1 receptor gene (IGF1r) mRNA in blastocysts. This study proposes that this flavonoid could increase mRNA IGF1r and IGF1 in the reproductive system. The protective role of quercetin is associated with increased estrogen receptor expression in the different stages of preimplantation in mice embryos. Furthermore, quercetin increased mRNA expression levels of integrin αv and β3 subunits and cyclooxygenase-2 (COX-2) mRNA in blastocysts. This flavonol decreased the expression of Caspase3 mRNA levels, an indicator of apoptosis, and activated the Wnt-β-catenin nuclear signaling pathway, which acts on blastocyst activation during endometrial implantation. Therefore, this research infers that GDM is related to low estrogen levels in early pregnancy, altered mRNA expression of the IGF1r, integrin αvβ3, and COX-2 genes, and inhibits the activation of the Wnt-β-catenin pathway. Conversely, supplementation with quercetin during perinatal life may relieve these adverse effects, highlighting the therapeutic potential of this flavonol on GDM management [40].

## 5. Quercetin Intervention in Gestational Undernutrition

### 5.1. Gestational Undernutrition: A Pregnant Disorder Linked to Oxidative Stress and Inflammation

Maternal malnutrition is associated with fetal development delay and damage to several organs, including brain and skeletal muscle [42,43]. Regarding neurological integrity, exposure to perinatal protein malnutrition results in smaller brain size, delayed ontogenesis of reflexes, change in neural circuits responsible for hunger and satiety patterns, and reduced cognitive functions. Moreover, this insult significantly decreased body weight, muscle mass, and metabolism, causing damage to locomotor activity [44,45]. Structural changes are also observed in orofacial muscles and neural circuits related to mastication, altering the pattern of masticatory cycles and decreasing mastication efficiency [46].

Protein restriction in early life impairs lipid and glucose metabolism, increasing the predisposition to obesity and cardio-metabolic diseases throughout life [47,48,49,50]. In addition, perinatal protein restriction leads to decreased insulin and leptin sensitivity, increased body adiposity, decreased satiety, and reduced basal energy expenditure [51]. Putative mechanisms associated with chronic diseases after undernutrition in early life are related to increased oxidative stress and inflammation caused by the lack of essential nutrients, such as essential amino acids [42,51,52,53].

Once perinatal protein restriction increases the concentration of free radicals and inflammatory cytokines, experimental evidence investigates the administration of antioxidant and anti-inflammatory bioactive compounds on the phenotypic plasticity of malnourished animals in early life [10,54,55]. Among these compounds, quercetin stands out due to its antiobesity, anti-inflammatory, antioxidant and antidyslipidemic functions, which may contribute to regulating body weight, biochemical, and metabolism parameters [56].

### 5.2. Quercetin Supplementation in Rodent Models of Gestational Undernutrition

Animal evidence shows that quercetin supplementation can reduce the impact of gestational undernutrition in dams and rodent offspring (Table 2). The effects of quercetin during pregnancy on litter size and late-onset puberty induced by prenatal and postnatal food deprivation have been early evaluated in rats. The study performed three experiments: in the first experiment, rats were subjected to undernutrition during pregnancy (50% quantitative and qualitative reduction of casein pellets), and quercetin was administered at 50, 100, and 200 mg/kg. In the second experiment, undernutrition was induced by an increased number of pups per cage, and quercetin was administered at the same doses as in the first experiment during lactation. In the third experiment, the combination of the two insults induced undernutrition, and quercetin was administered from early pregnancy to weaning. The results showed that administration of a high dose (200 mg/kg) of quercetin significantly reduced the number of stillbirths induced by perinatal protein restriction and increased the body weight of the pups as well as reduced the number of stillbirths. A reduction in the incidence of late puberty was observed in malnourished pups after administration of quercetin [10].

Using the same study design protocol, the effect of quercetin was evaluated on prenatal and early postnatal food deprivation-induced changes in maternal and neonatal rat offspring outcomes. Treated quercetin animals showed an increase in both maternal and neonatal pup body weight. Additionally, quercetin modulated adulthood parameters, preventing leptin suppression, regulating ghrelin expression, and brain oxidative stress in the malnourished group. Quercetin limited body weight gain in adult rats subjected to perinatal protein restriction. One mechanism explaining these results was that quercetin increased leptin (anorexigenic hormone) and reduced ghrelin expression (orexigenic hormone), reducing eating and increasing energy expenditure. Furthermore, the study showed that quercetin mitigated perinatal malnutrition-induced oxidative stress in the rat brain by increasing glutathione (GSH) concentrations [54].

Another action of quercetin under investigation is its effect on AMP-activated protein kinase (AMPK). The consequences of perinatal malnutrition on the expression of AMPK in the liver of adult rats, as well as the administration of quercetin on the expression of this protein, have been early investigated [55]. Perinatal protein undernutrition was replicated by a diet containing 8% casein. From birth until week 23 of postnatal life, the control group received a standard chow diet, and treated animals received a diet enriched with 0.2% quercetin. The group supplemented by quercetin showed similar results to the control group, with higher body weight than the malnourished group in adulthood. In addition, quercetin increased the activation of AMPK in the adult offspring of restricted protein dams and modulated the AMPK pathway in the liver. Furthermore, malnourished rats that received quercetin had higher serum levels of insulin and lower levels of triglycerides (TG) compared to untreated diabetic rats [55]. In summary, rodent data demonstrated quercetin’s therapeutic potential in relieving damage triggered by perinatal protein restriction; however, the role of quercetin needs to be explored in clinical studies to confirm its effectiveness in malnourished patients.

## 6. Quercetin Supplementation in High-Fat Diet Rodent Models Inducing Maternal Overweight and Dyslipidemia

Maternal obesity is one of the most relevant health issues during pregnancy. Maternal obesity during gestation and lactation can increase the risk of pregnancy cardio-metabolic disorders, including GDM and pregnancy-induced hypertension [57]. Furthermore, clinical and animal studies have demonstrated that gestational obesity is associated with an increased maternal and infant mortality rate and predisposes offspring to develop obesity and other related metabolic dysfunctions in adulthood [58,59].

Commonly, maternal obesity results from nutritional inadequacies characterized by the accentuated consumption of high-calorie and high-fat processed foods. In this sense, a range of rodent models has been used to replicate obesity through high-fat diet (HFD) consumption and investigate the role of bioactive compounds in gestational obesity management [11,59,60,61,62]. The current treatments for gestational obesity are based on lifestyle modifications and drug consumption, which have limited effectiveness [63]. Thus, particular focus has been given to natural compounds, widely found in fruits and vegetables, once they can regulate physical, metabolic, and biochemical parameters, as quercetin does.

Quercetin has been investigated as a potential molecule for combatting obesity (Table 3) due to its antiobesity activity, antioxidant and anti-inflammatory properties, which could improve the uteroplacental unit of the obese mother [11,59]. Some clinical studies demonstrated that daily consumption of quercetin could be beneficial for preventing obesity-related complications in adulthood [64,65]. However, it remains unclear if this beneficial effect may be observed in obese pregnant, which limits the recommendation of quercetin during gestation and lactation.

Few preclinical studies investigated the role of quercetin during the perinatal period in HFD animals, and the results are controversial. Quercetin supplementation (150 mg/kg) from the beginning of gestation to gestational day 19 promoted little effect on physical and biochemical parameters in rats exposed to HFD (which contained 45% fat) [60]. According to this study, treatment with quercetin could attenuate MDA, SOD, and nitric oxide (NO) concentration in the placenta and liver tissue of neonate rats. Similarly, another study analyzed the same method of supplementation of quercetin reduced expression of mRNA of TNF-α and interleukin-1B (IL-1B) in the placenta and hypothalamus of neonatal rats, as well as an increase in NF-κB mRNA level in placenta [61]. It was highlighted that 45% of fat applied in this study did not provoke severe metabolic changes, which suggests that future studies could consider using a higher percentage of fat to improve the quality of the HFD model to mimic changes induced by obesity [60].

Another experimental study evaluated the long-term effects of quercetin in a dosage of 66 mg/kg for 4 months in mice exposed to HFD with a higher fat percentage (60%). HFD-induced mice developed hyperglycemia, hyperinsulinemia, and insulin resistance and exhibited evidence of accelerated aging. On the other hand, treatment with quercetin reduced blood pressure, blood glucose, and plasma insulin levels in 6- and 12-month mice offspring [62].

Recently, it was demonstrated that prolonged administration of quercetin changed physical and metabolic parameters in rats subjected to HFD with a lower percentage of fat (30%). This study revealed that quercetin induced minor effects, including elevation of body and liver weight of dams exposed to HFD, and also changed the lipid profile of neonatal mice. However, the 30% fat used in this study may not effectively replicate physical and metabolic damage observed in gestational obesity [11].

It has been observed that a high-fat diet can induce dyslipidemia. In this sense, two of the previously mentioned studies analyzed the effects of quercetin as a protector of changes caused by HFD in lipid parameters [11,59]. Including quercetin at 1.0% in a diet could help regulate the low-density lipoprotein cholesterol (LDL-C) and high-density lipoprotein cholesterol (HDL-C). However, maternal consumption of quercetin did not affect body weight or blood lipid parameters in either dams or neonates at postnatal day 3 [11]. Another study demonstrated that 50, 100, or 200 mg/kg of quercetin during gestation and lactation contributed to the regulation of lipid parameters after deregulation induced for HFD. In this context, significant changes were found in maternal TG, cholesterol, and HDL-C as gestation progressed, with attenuation due to quercetin supplementation, especially at higher doses [59]. Additionally, quercetin supplementation triggered long-term effects, including reduced production of pro-inflammatory markers (TNF-α, IL-6), decreased endoplasmic reticulum stress, NF-κB protein expression, and downregulating the c-Jun N-terminal kinase signaling pathway (p-JNK). In summary, the treatment with quercetin improved a range of physical, metabolic, and biochemical parameters. However, there is a lack of information on these protective effects in randomized clinical trials (RCTs) with women affected by gestational obesity.

## 7. Quercetin Supplementation in Rodent Models of Pregnancy Hypertension Syndromes

Hypertensive disorders during pregnancy include a range of maternal disorders, such as chronic hypertension, preeclampsia, and gestational hypertension. All these conditions are commonly called pregnancy-induced hypertension [13]. Among these disorders, preeclampsia is estimated to affect 5–8% of pregnancies annually [12]. In addition, preeclampsia is considered one of the more prevalent causes of maternal and fetal mortality, with more than 76,000 maternal and 500,000 fetal deaths annually worldwide [66,67]. The clinical characteristics of this disorder often appear after the 20th week of gestation, including maternal hypertension and proteinuria [13,67].

The pathophysiological mechanisms include an imbalance between placental vasoconstrictors and vasodilators, with a reduction of NO, one of the primary regulators of the blood flow in the uteroplacental unit [13]. Due to deregulation between vasoconstrictor and vasodilator agents, placental vascular resistance increases, leading to cellular hypoxia and ischemia, enhanced oxidative stress, and lipid peroxidation [23]. Coupled with an elevation of oxidative stress, increased maternal systemic inflammatory response resulting from high serum and placental pro-inflammatory cytokine levels play an essential role in the development of preeclampsia [23].

Numerous animal models are utilized to mimic signs and symptoms of preeclampsia, including exposure to LPS [12], reduced uterine perfusion pressure [68], and inhibition of nitric oxide synthase (NOS) agents [23,67]. According to experimental findings, most animal models of preeclampsia could replicate elevation of systolic and diastolic blood pressure, proteinuria, increased oxidative stress and proinflammatory markers, and the appearance of adverse fetal outcomes. The replication of these animal models can be used to investigate new and safe therapeutic strategies that may alleviate preeclampsia-related symptoms. Considering quercetin’s antioxidant and anti-inflammatory properties, findings have shown that quercetin supplementation may improve clinical management of pregnancy-induced hypertension (Table 4).

Experimental studies using gravid rats revealed that treatment with quercetin could not induce placental maladaptation or teratogenic effects on the fetus, highlighting the potential safety of this bioactive compound [13].

An experimental study evaluated the efficacy of quercetin supplementation in rats subjected to preeclampsia by exposure to LPS [12]. The supplementation with quercetin improved maternal outcomes, including reduced systolic blood pressure and proteinuria. Quercetin administration triggered a protective placental response, observed by reversal of imbalance of angiogenic factors, reduced placental expression of TNF-α, IL-6, monocyte chemoattractant protein-1 (MCP-1), MDA, and increased placental weight. Coupled with maternal and placental findings, groups treated with quercetin had improved fetal and neonatal outcomes, such as reduced reabsorbed fetuses and increased pups weight. However, this study did not mention the dose and duration of treatment, limiting the interpretation of these findings.

The reduced uterine perfusion pressure rat model was used to explore the effect of quercetin on hypertension during pregnancy [68]. Quercetin supplementation at 10, 20, or 50 mg/kg offered from gestational day 14 to day 21 exerted beneficial effects on dams, placental, and fetus. In dams, quercetin supplementation reduced systolic and diastolic blood pressure and decreased endothelin-1 (ET-1) serum levels and pro-inflammatory cytokines TNF-α and IL-6. ET-1 is expressed by smooth muscle cells and vascular endothelial cells, and the overexpression of ET-1 results in high blood pressure in preeclamptic patients. Quercetin treatment reduced gene and protein expression of ET-1 and ET_A_R in placenta tissue. In addition, quercetin treatment reduced fetal resorptions and increased fetal body weight [68].

Another experimental study investigated a preeclampsia rat model using an inhibitor of NOS with NG-nitro-Larginine-methyl-ester (L-NAME) [23]. A single administration of quercetin (10 mg/kg) at the 17th gestational day reduced maternal plasma MDA, CAT, and SOD levels, and improved renal function by reducing proteinuria. Quercetin administration improved neonatal survival through higher percentages of live-born pups and lower rates of dead pups. Despite these findings, no changes were observed in maternal blood pressure and birth weight. Another experimental study that used the same preeclampsia induction method verified that administration of quercetin (2 mg/kg) from the 4th to 19th gestational day induced minor effects on maternal, placental, and neonatal outcomes. According to this study, quercetin could regulate proteinuria and promote antioxidant and anti-inflammatory effects [67]. However, quercetin potentiated the positive effects induced by the administration of aspirin, suggesting that quercetin may be effective as an adjunct therapy. These experimental findings recommended that quercetin supplementation may attenuate maternal signs of preeclampsia and improve a range of placental, fetal, and neonatal outcomes.

Results of RCTs showed that administration of quercetin could lower both systolic and diastolic blood pressure of hypertensive patients [69,70,71]. However, little is known about the efficacy and safety of quercetin in attenuating maternal and fetal outcomes resulting from preeclampsia or other gestational hypertensive disorders. Therefore, quercetin’s safety in treating pregnant women with preeclampsia needs to be further explored in RCTs.

## 8. Conclusions and Future Perspectives

Studies in rodent models demonstrated the potential of quercetin supplementation during pregnancy and lactation in attenuating signs and symptoms of cardio-metabolic diseases, including GDM, dyslipidemia, hypertensive syndrome, and causative disorders, such as undernutrition and maternal overweight (Figure 1). Experimental evidence revealed that gestational cardio-metabolic diseases expose the embryo, fetus, and neonates to a pro-inflammatory and high oxidative stress environment through uteroplacental unit or breastfeeding. Conversely, according to studies performed with rodents, a reduced proinflammatory state and decreased oxidative and nitrosative stresses could be considered one of the putative mechanisms by which maternal quercetin consumption could alleviate damage induced by gestational cardio-metabolic diseases on their progeny.

The available evidence performed with rodent models indicates that the beneficial effects of maternal quercetin supplementation could result from alterations in epigenetic mechanisms. It could be proposed that epigenetic changes may stimulate genes related to the AMPK, NF-κB, and phosphoinositide 3-kinase (PI3K)/AKT pathways, although further studies are required to understand these mechanisms. Histone and DNA methyltransferases alterations establish AMPK as a key player in epigenetic regulation. For instance, metabolic disorders, such as maternal obesity are often associated with reduced AMPK expression, which may be relieved by quercetin supplementation [72]. In addition, quercetin supplementation could stimulate the AMPK pathway, which attenuates the expression of inflammatory mediator genes through NF-κB suppression [73]. Additionally, the activation of the AMPK pathway can lead to inhibition of acetyl-CoA carboxylase and enhanced expression of endothelial nitric oxide synthase (eNOS) [55]. Furthermore, this bioactive compound may stimulate the expression and translocation of glucose transporter type 4 (GLUT4) and activate the mechanistic target of rapamycin (mTOR) via PI3K activation [73].

Lastly, it is important to highlight that the heterogeneities of studies, including the variation of duration, diverse range of doses, and co-interventions, limit the understanding of quercetin’s efficacy in managing maternal cardio-metabolic disorders. In non-pregnant human subjects, systematic reviews and meta-analyses of randomized, controlled trials assessing the effects of quercetin in a range from 30 mg/d to 1000 mg/d have reported beneficial effects on cardio-metabolic parameters [74,75]. Additionally, randomized clinical trials have demonstrated that quercetin at a daily dose of 1000 mg is safely tolerated in patients with chronic obstructive pulmonary disease [76] and COVID-19 [77]. Although randomized and controlled clinical trials have not been performed with pregnant women, it is reasonable to suggest that quercetin-rich fruits and vegetables should be inserted into daily dietetic prescription during pregnancy (Figure 2). It is essential to mention that once all findings assessing the effects of quercetin supplementation during pregnancy came from studies in rodent, future studies exploring the optimal dosage and intervention duration in humans could help accelerate the application in pregnant women.

## Figures and Tables

**Figure 1 foods-11-02772-f001:**
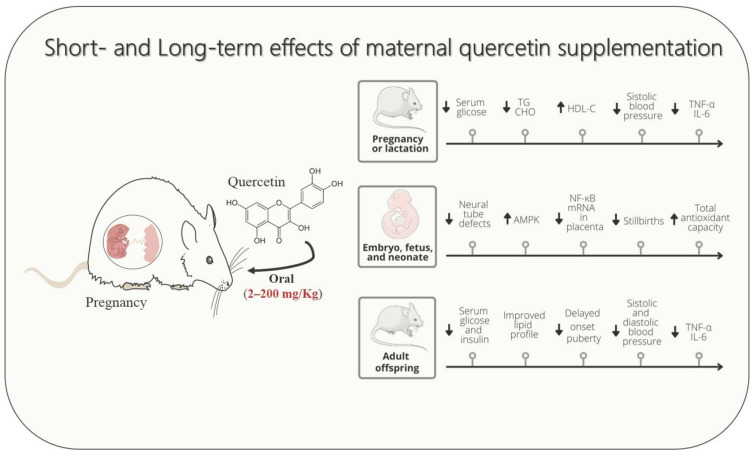
Schematic drawing showing that maternal quercetin supplementation improves cardio-metabolic, inflammatory, and oxidative stress parameters in dams, fetuses, and offspring. The up arrow ↑ represents increase, down arrow ↓ represents decrease.

**Figure 2 foods-11-02772-f002:**
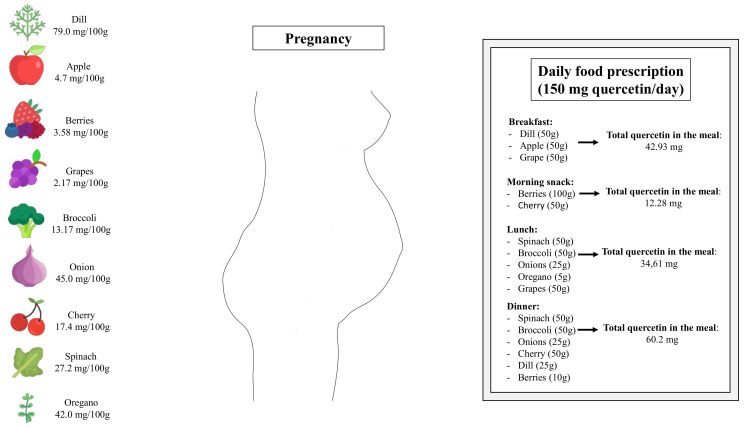
Schematic drawing showing sources of quercetin-rich fruits and vegetables and how the presence of certain foods can be essential in a planned food prescription during pregnancy.

**Table 1 foods-11-02772-t001:** Studies using quercetin in the treatment of diabetes mellitus during pregnancy. The up arrow ↑ represents increase, down arrow ↓ represents decrease.

Reference	Type of Study	Dose	Duration	Maternal Outcomes	Embryonic, Fetal, and Neonatal Outcomes
Cao et al., 2016 [24]	Experimental (mice)	100 mg/kg	Embryonic period: 7.5 to 10.5.	No difference in maternal glucose.	↓ Decreased the neural tube defects rate; ↓ Apoptosis in the dorsal neural tube of the embryos and levels of Caspase-3; ↓ Nos2 level in the neural tissues of the embryos; ↓ Levels of protein S nitrosylation in the neural tube; ↑ Enzyme levels of SOD1 and SOD2; ↓ Levels of 4-HNE and MDA; ↑ Expression of redox regulating and DNA damage; There was the presence of quercetin metabolites in the embryo.
Tan et al., 2018 [39]	Experimental (mice)	100 mg/kg	Embryonic period: 6.5 to 9.5.	The authors did not evaluate maternal outcomes.	↓ Neural tube defects; ↓ The apoptotic signals were lower in the neural tube regions; ↓ Nos2 expression in the embryos. ↑ The levels of Sod1 in the embryos; ↓ Nitrosative and oxidative stresses in the endoplasmic reticulum (ER); ↓ The expression of p65.
Bolouki et al., 2020 [40]	Experimental (mice)	30 mg/kg	4 weeks before conception.	↓ Levels of blood glucose; ↑ The number of embryos per mouse; ↑ Levels of serum 17β-estradiol; ↑ Estradiol/progesterone ratio.	↑ The embryo morphological distribution to the well-developed stages; ↑ IGF1r, integrin αvβ3, and Cox2 mRNA express these genes in the blastocyst; ↓ Expression of the Caspase3 gene; Can activate the nuclear Wnt-β-catenin signaling pathway.
Mahabady et al., 2021 [41]	Experimental (rats)	75 mg/kg	On 0, 7, 14, and 20 days of gestation.	The authors did not evaluate maternal outcomes.	↓ LIM thickness; ↓ Mean the number of glycogen cells; ↑ Increased placental adiponectin expression; ↓ Placental expression of AdipoR1 and AdipoR2.

**Table 2 foods-11-02772-t002:** Studies using quercetin in the treatment of perinatal undernutrition. The up arrow ↑ represents increase, down arrow ↓ represents decrease.

Reference	Type of Study	Dose	Duration	Maternal Outcomes	Embryonic, Fetal, and Neonatal Outcomes	Adulthood Outcomes
Anachuna et al., 2020 [10]	Experimental (rats)	50, 100 and 200 mg/kg	21–22 days—during pregnancy e/or 22 days during weaning.	The authors did not evaluate maternal outcomes.	↓ Stillbirths; ↑ Nose-tail lenghts at P1 and P22.	↓ Onset puberty.
Anachuna et al., 2020 [54]	Experimental (rats)	50, 100 and 200 mg/kg	21–22 days—during pregnancy e/or 22 days during weaning.	↑ Maternal weight.	↑ Body weight undernourished rats until weaning	↓ Body weight; ↑ Leptin levels; ↓ Ghrelin levels; ↓ Brain oxidative stress.
Sato et al., 2013 [55]	Experimental (rats)	80–110 mg/day	10 to 22 postnatal days.	The authors did not evaluate maternal outcomes.	↓ TG blood level; ↑ AMPK.	↑ Body weight; ↑ Adiponectin; ↑ AMPK.

**Table 3 foods-11-02772-t003:** Studies using quercetin in the treatment on HDF models inducing maternal overweight and dyslipidemia. The up arrow ↑ represents increase, down arrow ↓ represents decrease.

Reference	Type of Study	Dose	Duration	Maternal Outcomes	Embryonic, Fetal and Neonatal Outcomes	Adulthood Outcomes
Adeyemi et al., 2021 [60]	Experimental (rats subjected to HFD contained 45% of fat)	150 mg/kg	From the beginning of gestation to gestational day 19.	The authors did not evaluate maternal outcomes.	↓ MDA and NO concentration in the placenta and liver tissues in neonate male rats; ↓ SOD concentration in the placenta and liver tissues in neonatal rats; ↑ Total antioxidant capacity of the liver in neonates.	The authors did not evaluate adulthood outcomes.
Adeyemi et al., 2021 [61]	Experimental (rats subjected to HFD contained 45% of fat)	150 mg/kg	From the beginning of gestation to gestational day 19.	The authors did not evaluate maternal outcomes.	↓ Expression of mRNA of TNF-α and IL-1β in placenta and hypothalamus of neonatal rats; ↑ NF-κB mRNA level in placenta.	The authors did not evaluate adulthood outcomes.
Liang et al., 2009 [62]	Experimental (mice subjected to HFD contained 60% of fat)	66 mg/kg	For 4 weeks before breeding.	The authors did not evaluate maternal outcomes.	The authors did not evaluate embryonic, fetal, and neonatal outcomes.	↓ Blood glucose and plasma insulin levels of 6- and 12-month offspring; ↓ Systolic and diastolic blood pressure of 6- and 12-month offspring.
Takashima et al., 2021 [11]	Experimental (mice subjected to HFD contained 30% of fat)	1.0%	Before breeding, throughout gestation, lactation until 13 weeks of postnatal days.	↑ Body weight; ↑ Liver weight.	↑ Total CHO, non-HDL-C, and HDL-C levels of pups; ↓ Gastric inhibitory polypeptide levels of pups.	The authors did not evaluate adulthood outcomes.
Wu et al., 2014 [59]	Experimental (rats subjected to HFD contained about 42% of fat)	50, 100 or 200 mg/kg	Throughout gestation and lactation.	Did not change body weight and blood glucose levels; Improved lipid profile (↓ TG, CHO, and ↑ HDL-C levels).	The authors did not evaluate embryonic, fetal, and neonatal outcomes.	↓ Blood glucose and insulin levels in adult rats; Improved lipid profile (↓ TG, CHO) in adult rats; ↓ Expression of TNF-α, IL-6 in adult rats; ↓ Endoplasmic reticulum stress in liver and adipose tissues; ↓ In p-JNK and NF-κB protein expression.

**Table 4 foods-11-02772-t004:** Studies using quercetin in the treatment of models of complicated pregnancies, including preeclampsia. The up arrow ↑ represents increase, down arrow ↓ represents decrease.

Reference	Type of Study	Dose	Duration	Maternal Outcomes	Embryonic, Fetal and Neonatal Outcomes
Li et al., 2020 [12]	Experimental (rats subjected to preeclampsia model)	The authors did not inform the dose of treatment.	The authors did not inform the duration of treatment.	↓ Systolic blood pressure; ↓ Proteinuria.	Reversed imbalance of angiogenic factors production in the placenta; ↓ Placental growth factor; ↓ Placental expression of TNF-α, IL-6, and MCP-1 levels; ↓ Placental MDA; ↑ Placenta weight; ↓ Reabsorbed fetuses %; ↑ Pups weight; ↑ Placenta weight.
Sun et al., 2020 [68]	Experimental (rats subjected to gestational hypertension)	10 mg/kg, 20 mg/kg, or 50 mg/kg.	From gestational day 14 to gestational day 21.	↓ Systolic and diastolic blood pressure; ↓ Plasma level of ET-1, sFLT1; ↑ Plasma level of VEGF; ↓ Plasma levels of TNF-α and IL-6, and ↑ IL-10.	↓ ET-1, ET_A_R expression in placenta tissue; ↑ Fetal weight; Did not change placental weight; ↑ Fetal weight/ Placenta weight; ↓ Reabsorbed fetuses %.
Tanir et al., 2005 [23]	Experimental (rats subjected to preeclampsia model)	10 mg/kg	Single administration at gestational day 17.	Did not change blood pressure;↓ Plasma MDA levels; ↓ Erythrocyte CAT and SOD levels; ↓ Proteinuria.	Did not change the birth weight of pups; ↑ Neonatal survival (higher percentages of liveborn pups and lower rates of dead pups).
Yang et al., 2019 [67]	Experimental (rats subjected to preeclampsia model)	2 mg/kg	From gestational day 4 to gestational day 19.	↓ Proteinuria; ↓ Plasma MDA levels; Did not change TNF-α; ↓ Plasma levels of IL-6.	Did not change placental MDA; Did not change placental expression of TNF-α and IL-6; Did not change relative mRNA expression of uterus VEGF and sFlt-1.

## Data Availability

Data available on publication. The data used to support the findings of this study can be made available by the corresponding author upon request.

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
