# Peer review of "Evidence for Quercetin as a Dietary Supplement for the Treatment of Cardio-Metabolic Diseases in Pregnancy: A Review in Rodent Models"

_foods, 2022, doi:10.3390/foods11182772_

Round 1
Reviewer 1 Report
Reviewer's comment on Manuscript Number: foods-1815707
The manuscript entitled “Efficacy of quercetin as a dietary supplement for the treatment of cardio-metabolic diseases in pregnancy: a review” falls within the scope of Foods.
The manuscript is certainly very interesting and valuable. However, I have a few remarks:
- Tables should be cited in the manuscript.
- There should be added additional subheadings in the manuscript. Now, it is challenging to read a part that for instance is described on three pages (see points 4 and 5 and more). Such division would allow readers to understand the text better.
- It would be good to add graphical abstract to the paper.
- Why the submitted paper was not prepared on the template?
- I wonder if it is justified to use the phrase "dietary supplement" in the title since the work did not describe the usage of a particular supplement but supplementation of the diet with quercetin – I recommend the change in the title and throughout the manuscript
- Please check the English quality of the paper. It needs proofreading.
Author Response
We thank the reviewers for their valuable comments, which were very important to getting a publishable article. We have carefully addressed the significant points, as discussed below. We hope the revised version will meet the requirements for being published in Foods.
Reviewer: 1
The manuscript entitled “Efficacy of quercetin as a dietary supplement for the treatment of cardio-metabolic diseases in pregnancy: a review” falls within the scope of Foods.
The manuscript is certainly very interesting and valuable. However, I have a few remarks:
Author response: We would like to thank reviewer 1 for recognizing the relevance of our work and the opportunity to improve our manuscript. We have made the adjustments proposed by the reviewer, which contributed to improving the manuscript's organization
- Tables should be cited in the manuscript.
Author response: We thank the reviewer for this attention. We’ve inserted in the text the citation of the tables. We’ve highlighted the modifications in red. Please, see lines 244 (Table 1), 377 (Table 2), 449 (Table 3), and 543 (Table 4).
- There should be added additional subheadings in the manuscript. Now, it is challenging to read a part that for instance is described on three pages (see points 4 and 5 and more). Such division would allow readers to understand the text better.
Author response: We thank the careful reading made by the reviewer. We’ve added a content section to improve readers' understanding. Please, see lines 45-54.
- It would be good to add graphical abstract to the paper.
Author response: We thank you for the suggestion. We added a graphical abstract to the paper (line 706).
- Why the submitted paper was not prepared on the template?
Author response: Sorry, reviewer 1. It was a mistake.
- I wonder if it is justified to use the phrase "dietary supplement" in the title since the work did not describe the usage of a particular supplement but supplementation of the diet with quercetin – I recommend the change in the title and throughout the manuscript
Author response: We thank the reviewer for carefully reading the paper. Considering that the studies used quercetin as a manipulated/marked supplement and not included in diet/food, we decided to maintain the term dietary supplement.
- Please check the English quality of the paper. It needs proofreading.
Author response: We thank the reviewer for the comment. We reviewed the English carefully to improve the writing quality. We’ve highlighted the modifications in red.
Reviewer 2 Report
Review of Foods 1815707
My limited experience with reviewing reviews is that they usually describe their criteria for including papers and use some sort of checklist. This paper does neither.
Since I know nothing about this compound, I did an internet search. This found me plenty of information both pro and con at reputable sites (Mt. Sinai New York, Mayo Clinic, Wikipedia). Also, I’m a statistician, not a food scientist, so I am taking on faith that the authors are summarizing the papers reviewed fairly, and that the papers represent competently done research, though many seem to be in super-obscure journals.
All the papers mentioned are based on animal (mostly or all rodents, I think) studies. Yet the conclusions seem to be happy to jump to clinical trials in pregnant women. It doesn’t seem as if there are even any serious trials in nonpregnant humans, though there seems to be some observational data. I note that lab animals live restricted lives, while humans have varied diets and often use medications that may interact with quercetin. The ethics of doing trials in pregnant women at this time seem dubious to me, but there may be a role for observational studies. At the very least, the authors need to tone down their conclusions and suggestions.
Author Response
We thank the reviewers for their valuable comments, which were very important to getting a publishable article. We have carefully addressed the significant points, as discussed below. We hope the revised version will meet the requirements for being published in Foods.
Reviewer: 2
- My limited experience with reviewing reviews is that they usually describe their criteria for including papers and use some sort of checklist. This paper does neither.
Author response: We appreciate the reviewer's comments. We understand that this review model used by the authors represents the best alternative at the moment. A systematic review, as I believe the reviewer is referring to, could be carried out at another time, especially when more clinical data are available.
- Since I know nothing about this compound, I did an internet search. This found me plenty of information both pro and con at reputable sites (Mt. Sinai New York, Mayo Clinic, Wikipedia). Also, I’m a statistician, not a food scientist, so I am taking on faith that the authors are summarizing the papers reviewed fairly, and that the papers represent competently done research, though many seem to be in super-obscure journals.
Author response: We thank the reviewer for the comment. We believe that the journals used in the references express relevant and reliable information for this review, as they are indexed in relevant databases and have good metrics.
- All the papers mentioned are based on animal (mostly or all rodents, I think) studies. Yet the conclusions seem to be happy to jump to clinical trials in pregnant women. It doesn’t seem as if there are even any serious trials in nonpregnant humans, though there seems to be some observational data. I note that lab animals live restricted lives, while humans have varied diets and often use medications that may interact with quercetin. The ethics of doing trials in pregnant women at this time seem dubious to me, but there may be a role for observational studies. At the very least, the authors need to tone down their conclusions and suggestions.
Author response: We agree with the reviewer that we have to soften this information. We re-write the sentence as following: “It is essential to mention that once all findings came from preclinical studies, future studies exploring the optimal dosage and intervention duration in humans could help accelerate the application in pregnant women.”
Round 2
Reviewer 1 Report
Reviewer's comment on Manuscript Number: foods-1815707
The manuscript was revised by authors, but still some remarks were not addressed properly.
- There should be added additional subheadings in the manuscript. I meant to add subparagraphs in points 4 and 5. The content list is not necessary.
- The authors suggest in the abstract “Quercetin supplementation during pregnancy and lactation has been linked to a lower risk of maternal cardio-metabolic disorders such as gestational diabetes mellitus (GDM), dyslipidemia, preeclampsia, and attenuation of malnutrition-related conditions, and gestational obesity. “ – however they do not mention that whole review concern animal studies. It should be clearly stated both in the title and the abstract.
- For instance in conclusions” The available evidence indicates that the beneficial effects found in adulthood could result from permanent alterations in epigenetic
regulation and related phenotypes in rodents treated with quercetin.” – this sentence suggests that information concerns humans but is s not. Every statement concerning the efficacy of quercetin supplementation should be accompanied by details concerning what animal model was used. There is a great difference between animals and human models.
I propose to accept this paper for publication in Foods after minor amendments.
Author Response
The manuscript was revised by authors, but still some remarks were not addressed properly.
Author’s response: We would like to thank again reviewer 1 for recognizing the relevance of our work and the opportunity to improve our manuscript. We have made the adjustments proposed by the reviewer, which contributed to improving the manuscript's organization.
There should be added additional subheadings in the manuscript. I meant to add subparagraphs in points 4 and 5. The content list is not necessary.
Author’s response: We’ve removed the content list in the revised version and added additional subheadings in points 4 and 5 as requested by reviewer 1.
The authors suggest in the abstract “Quercetin supplementation during pregnancy and lactation has been linked to a lower risk of maternal cardio-metabolic disorders such as gestational diabetes mellitus (GDM), dyslipidemia, preeclampsia, and attenuation of malnutrition-related conditions, and gestational obesity. “ – however they do not mention that whole review concern animal studies. It should be clearly stated both in the title and the abstract.
Author’s response: We agree with reviewer 1. We’ve highlighted in the title and abstract of the revised version the concern addressed by reviewer 1.
For instance in conclusions” The available evidence indicates that the beneficial effects found in adulthood could result from permanent alterations in epigenetic
regulation and related phenotypes in rodents treated with quercetin.” – this sentence suggests that information concerns humans but is s not. Every statement concerning the efficacy of quercetin supplementation should be accompanied by details concerning what animal model was used. There is a great difference between animals and human models.
Author’s response: Thank you reviewer 1. In revised version we’ve corrected the sentence.
I propose to accept this paper for publication in Foods after minor amendments.
Author’s response: Thank you reviewer 1.
Reviewer 2 Report
Review of Foods-1815707.v2
The manuscript is improved. The tables are well done and helpful. I know they were there before, but I did not pay much attention to them the first time.
Does “preclinical” always mean animal experiments? If not, they should be more precise.
The authors should make clear in the prose what animals were used in the experimental studies. They usually do, but not always. The uses of “dams” and “pups,” as opposed to “mothers” and “children” also imply animals as opposed to humans. The authors should perhaps avoid using “offspring,” which could mean either. Indeed, wherever possible, note that all the experimental data are from animals, as in line77-78, which I would rephrase as “Experimental studies in animal, as well as epidemiological studies, have….” This also applies to “children” in line 62 (in addition to the fact that the wording is noun, adjective, noun, which is awkward)
Starting with section 3, a lot of paragraphs seem to begin with generalizations about biochemistry, physiology, and/or quercetin, after which animal studies are cited. Please make sure that all effects mentioned are specified as to human or rodent. In fact, I would like to see all the section titles note the sources of the data.
Should more attention be given to publications showing detrimental effects (observational) in humans?
Line 242 Add “Animal” before “evidence.”
I am glad to see that the recommendation for clinical studies in pregnant women has been removed. If any trials are to be done, it would seem that provision of foods high in quercetin would be a good first step. Since all the foods high in quercetin are on the “good” list of the nutritionists, it would seem to be ethical to do extra provision of these foods in an experimental group (perhaps balanced with some other benefit or intervention, such as extra money/vouchers for food in the “control” group) of pregnant women, with the study including dietary evaluation in both groups. By the way, is it possible to evaluate quercetin intake based on blood or other measurements? If actual foods are used, “overdosing” is unlikely. But “underdosing” is possible.
A couple of small points:
On line 28, “chromatography” has an “h” after the “c” in English.
In line 105, “There is,” not “there are.”
Author Response
The manuscript is improved. The tables are well done and helpful. I know they were there before, but I did not pay much attention to them the first time.
Author’s response: We would like to thank again reviewer 2 for recognizing the relevance of our work and the opportunity to improve our manuscript. We have made the adjustments proposed by the reviewer, which contributed to improving the manuscript's organization.
Does “preclinical” always mean animal experiments? If not, they should be more precise.
Author’s response: Yes. However, to more precise information, we’ve changed the “preclinical” term to “animal” or “rodent”.
The authors should make clear in the prose what animals were used in the experimental studies. They usually do, but not always. The uses of “dams” and “pups,” as opposed to “mothers” and “children” also imply animals as opposed to humans. The authors should perhaps avoid using “offspring,” which could mean either. Indeed, wherever possible, note that all the experimental data are from animals, as in line77-78, which I would rephrase as “Experimental studies in animal, as well as epidemiological studies, have….” This also applies to “children” in line 62 (in addition to the fact that the wording is noun, adjective, noun, which is awkward)
Author’s response: We agree with reviewer 2. We’ve revised the manuscript to ensure information about the model used. Although we have used the term “offspring”, in each study we highlighted whether it was used mice or rat as models. In addition, in the revised version, we’ve removed the term “mothers” to study with animals. We’ve highlighted the modifications in yellow.
Starting with section 3, a lot of paragraphs seem to begin with generalizations about biochemistry, physiology, and/or quercetin, after which animal studies are cited. Please make sure that all effects mentioned are specified as to human or rodent. In fact, I would like to see all the section titles note the sources of the data.
Author’s response: We agree with reviewer 2. In the revised version we’ve assured that the findings are from rodent studies. We’ve highlighted in all the section titles that the findings are derived from animal studies. We’ve highlighted the modifications in yellow.
Should more attention be given to publications showing detrimental effects (observational) in humans?
Author’s response: We’ve revised clinical trials (not pregnancy) using quercetin supplementation. The studies have not reported toxic and harmful effects. Thus, detrimental effects were not reported in the manuscript.
Line 242 Add “Animal” before “evidence.”
Author’s response: We’ve added, as requested.
I am glad to see that the recommendation for clinical studies in pregnant women has been removed. If any trials are to be done, it would seem that provision of foods high in quercetin would be a good first step. Since all the foods high in quercetin are on the “good” list of the nutritionists, it would seem to be ethical to do extra provision of these foods in an experimental group (perhaps balanced with some other benefit or intervention, such as extra money/vouchers for food in the “control” group) of pregnant women, with the study including dietary evaluation in both groups. By the way, is it possible to evaluate quercetin intake based on blood or other measurements? If actual foods are used, “overdosing” is unlikely. But “underdosing” is possible.
Author’s response: Yes, quercetin may be measured in human plasma by a high-performance liquid chromatographic method (PMID: 9188819). Truly, we would like to thank reviewer 2 for the thought. This allows us to suggest that quercetin-rich foods should be part of a planned food prescription. Please see figure 2. Lastly, it is possible to estimate the quercetin intake through a 24-hour food recall.
A couple of small points:
On line 28, “chromatography” has an “h” after the “c” in English.
In line 105, “There is,” not “there are.”
Author’s Response: Thank you very much reviewer for the attention. We’ve corrected these mistakes.